# Association of Circadian Clock Gene Expression with Pediatric/Adolescent Asthma and Its Comorbidities

**DOI:** 10.3390/ijms24087477

**Published:** 2023-04-19

**Authors:** Nguyen Quoc Vuong Tran, Minh-Khang Le, Thuy-An Nguyen, Tetsuo Kondo, Atsuhito Nakao

**Affiliations:** 1Department of Immunology, Faculty of Medicine, University of Yamanashi, Yamanashi 409-3898, Japan; 2Department of Human Pathology, Faculty of Medicine, University of Yamanashi, Yamanashi 409-3898, Japanktetsuo@yamanashi.ac.jp (T.K.); 3Atopy Research Center, Juntendo University, School of Medicine, Tokyo 113-8421, Japan

**Keywords:** asthma, circadian clock, core clock genes, comorbidity, signaling pathways, allergic rhinitis, atopic dermatitis

## Abstract

The pathology of asthma is characterized by marked day–night variation, which is likely controlled by circadian clock activity. This study aimed to clarify the association of core circadian clock gene expression with clinical features of asthma. For this purpose, we accessed the National Center for Biotechnology Information database and analyzed transcriptomes of peripheral blood mononuclear cells and clinical characteristics of 134 pediatric/adolescent patients with asthma. Based on the expression patterns of seven core circadian clock genes (*CLOCK, BMAL1, PER1-3, CRY1-2*), we identified three circadian clusters (CCs) with distinct comorbidities and transcriptomic expressions. In the three CC subtypes, allergic rhinitis, and atopic dermatitis, both asthma comorbidities occurred in different proportions: CC1 had a high proportion of allergic rhinitis and atopic dermatitis; CC2 had a high proportion of atopic dermatitis but a low proportion of allergic rhinitis; and CC3 had a high proportion of allergic rhinitis but a low proportion of atopic dermatitis. This might be associated with the low activity of the FcεRI signaling pathway in CC2 and the cytokine–cytokine receptor interaction pathways in CC3. This is the first report to consider circadian clock gene expression in subcategories of patients with asthma and to explore their contribution to pathophysiology and comorbidity.

## 1. Introduction

Asthma is a chronic respiratory disorder characterized by airway inflammation and remodeling, with marked heterogeneity in its pathological features. It is a common illness among all age groups, with approximately 339 million affected people worldwide, particularly children (https://www.who.int/news-room/fact-sheets/detail/asthma, accessed on 23 January 2023). Interestingly, pediatric/adolescent asthma often co-exists with allergic rhinitis and atopic dermatitis, and together these conditions comprise an “allergic comorbidity cluster.” For instance, a study on more than 30,000 children between the ages of 4 and 8 years showed that children with asthma had significantly higher rates of comorbid allergic rhinitis and atopic dermatitis [1]. This tendency, however, is rarely seen in adults [2].

One of the distinguishing features of asthma is a marked day–night change in symptoms, airway physiology, and airway inflammation, typically with worsening at night. Recent studies suggest that diurnal variations in asthma are controlled by a body system that tracks the time of day, termed the circadian clock. For instance, the circadian clock contributes to the common nocturnal worsening of asthma, independent of sleep and other daily behavioral and environmental cycles [3]. Wild-type mice showed markedly different airway hyperresponsiveness (AHR) depending on the time of day, whereas the time-of-day effects on AHR were abolished in mice lacking the clock gene *Rev-erba*, in association with disruption of day–night changes in muscarinic receptors [4]. Thus, circadian clock activity can have a strong impact on asthma manifestations.

The mammalian circadian clock consists of interlocking transcription/translation feedback loops (TTFLs) of approximately 20 genes (“clock genes”) and acts as a molecular oscillator, turning on and off thousands of genes in a temporal manner within each cell [5]. The clocks are cell autonomous, but they can interact with daily light–dark and feeding-fasting cycles to synchronize their time with cycling environmental cues [6,7,8]. The core TTFL is that centered on the transcriptional factors CLOCK and BMAL1, which form a dimer, bind to an E-box sequence in the gene promoter region, and activate the transcription of a set of genes containing the E-box sequence. These genes include *Period1-3* (*PER1-3*) and *Cryptochrome 1-2* (*CRY1-2*). When activated by CLOCK/BMAL1, PER and CRY form a dimer in the cytoplasm, are translocated into the nucleus, bind to CLOCK/BMAL1, and inhibit its transcription activity. This negative feedback loop generates a 24-h rhythm with post-transcriptional modulations and confers a circadian rhythm on CLOCK/BMAL1 transcription activity. Consequently, the circadian rhythm also occurs in the expression of thousands of genes with the E-box sequence, and these genes are therefore referred to as clock-controlled genes as well as *PER* and *CRY* [9]. 

Although circadian clock activity can have a strong impact on asthma manifestations, few studies have addressed the association of circadian clock activity with clinical features of asthma. This study thus aimed to clarify the relationship between core circadian clock gene expression and manifestations of asthma. Using recently published transcriptome data of peripheral blood mononuclear cells (PBMCs) from individuals with pediatric/adolescent asthma [10], we found that these patients could be classified into three circadian cluster (CC) subtypes based on the expression patterns of seven core circadian genes (*CLOCK*, *BMAL1*, *PER1-3*, *CRY1-2*) that have transcriptomic distribution and were associated with different frequencies of two asthma comorbidities: allergic rhinitis and atopic dermatitis. This is the first report to consider the expression of circadian clock genes when subcategorizing patients with asthma and explore the association between clinical features and asthma-related signaling pathways in each subtype.

## 2. Results

### 2.1. Clustering Analysis Showed Three Distinct Core Circadian Clock Gene Expression Patterns in Patients with Pediatric/Adolescent Asthma

We first compared the read counts of seven core clock genes between control and asthma cohorts (Appendix A). Wilcoxon tests and the Bonferroni correction method were used. PER2 (adjusted *p* = 0.003) and PER3 (adjusted *p* = 0.002) expressions were statistically higher in the blood samples of asthma patients. We performed t-SNE dimension reduction of core clock gene expression, which can visually divide all samples into two clusters (*n* = 198 and *n* = 145) (Appendix A). A significant chi-square test (*p* < 0.001) was performed to illustrate the different distributions (52.5% in cluster 1 vs. 20.7% in cluster 2) of asthma patients between the two clusters. Therefore, there was an overall association between asthma and core clock gene expression patterns. 

With a focus on the asthma patients, the elbow method (Figure 1A) showed that significant drops in the within-cluster sums of squares (WSS) were present from K = 2 to K = 7, while the silhouette (Figure 1B) and gap statistic (Figure 1C) methods both indicated that K = 3 was the optimal value. These results showed that there were three distinct core circadian clock gene expression patterns (K = 3): circadian cluster (CC) 1 (*n* = 61), CC2 (*n* = 37), and CC3 (*n* = 36). A heatmap (Figure 1D) illustrates the differences in the expression of individual circadian clock genes. In the CC1 subtype, the *PER1, 2*, and *3* genes were expressed most highly, while these genes showed the lowest expression in the CC2 subtype. In addition, *CRY1* expression was high and that of *CRY2* was low in the CC1 subtype, and vice versa in CC2. On the other hand, *CLOCK* was more highly expressed in both CC2 and CC3 subtypes than in CC1, whereas its counterpart, *BMAL1*, was exclusively and intensively expressed in the CC3 subtype. Applying the same analysis workflow for the healthy control group resulted in two distinct CCs (Appendix A), suggesting that the three CC subtypes were characterized for asthma patients.

### 2.2. Core Circadian Clock Gene Expression Patterns May Be Related to Biological Differences in Pediatric/Adolescent Asthma

We performed t-SNE dimension reduction to demonstrate the distribution of the core circadian clock gene (Figure 1E) and transcriptomic expression (Figure 1F) hyperspaces. The circadian clusters were highly delineated in both hyperspaces, indicating the biological relevance of CC subtypes to circadian clock gene expression and transcriptomic programs in the patients’ blood. These observations were also found in the control groups (Appendix A), suggesting that biological differences between individuals may play important roles in shaping the transcriptomic profile. However, further exploration of the relationship between biological background, circadian clock, and transcriptomic profile in the control group is outside the scope of this study.

### 2.3. CC2 and CC3 Subtypes Were Characterized by a Low Frequency of Allergic Rhinitis and Atopic Dermatitis, Respectively

Table 1 summarizes the clinical information for the three CC asthma subtypes. Most of the patients with CC2 asthma were children (median 4.0; range 0.1–17.0 years), while adolescent patients were more common in the CC1 (median 16.8; range 3.9–18.0 years) and CC3 (median 16.8; range 4.5–17.8 years) subtypes. The difference in ages was significant (*p* < 0.001). The CC asthma subtypes were also differentially associated with comorbid allergic rhinitis and atopic dermatitis. Among CC1 patients, 27.9% and 23% exhibited allergic rhinitis and atopic dermatitis, respectively. These conditions occurred in 8.1% and 43.2% of CC2 patients and 44.4% and 5.6% of CC3 patients, respectively. Therefore, CC2 and CC3 had low proportions of allergic rhinitis and atopic dermatitis, respectively.

### 2.4. Kyoto Encyclopedia of Genes and Genomes (KEGG) Pathway Analysis Revealed Lower Activation of the FcεRI Receptor Signaling Pathway in CC2 and Lower Activation of Cytokine–Cytokine Receptor Interaction Pathways in CC3

We performed gene set variation analysis (GSVA) of all pathways in the Canonical Pathways gene sets in the “C2” category; these gene sets were derived from the KEGG pathway database (CP:KEGG) collection in the Human Molecular Signatures Database (MSigDB) database (msigdbr package version 7.5.1, https://CRAN.R-project.org/package=msigdbr, accessed on 9 December 2022). Multiple ANOVA tests were performed with Bonferroni correction to compare the pathway activities between the three CC subtypes (Appendix A). We also created a heatmap visualizing the signaling in the pathways of interest (Figure 2).

The activities of mitogen-activated protein kinase (MAPK), Janus kinase (JAK)/signal transducer and activator of transcription proteins (STAT), WNT, transforming growth factor-β (TGF-*β*), tight junctions, FcεRI, chemokines, insulin, and NOTCH signaling pathways were lower in the CC2 but higher in the CC1 and CC3 samples (adjusted *p* < 0.001). Lower signals of peroxisome proliferator-activated receptor (PPAR), calcium, and cytokine–cytokine pathways were seen in the CC3 and CC2 samples, while T-cell, B-cell, Toll-like, and nucleotide-binding oligomerization domain (NOD)-like signals were only high in the CC3 samples (adjusted *p* < 0.001).

## 3. Discussion

This study aimed to clarify the association of core circadian clock gene expression with asthma. Based on seven core circadian gene expression patterns, pediatric/adolescent patients with asthma were subdivided into 3 CC subtypes that had distinct comorbidities and transcriptomic expressions. CC1, characterized by a high expression of *PER* genes and low expression of *CLOCK* and *BMAL1*, had the highest prevalence of comorbid allergic rhinitis and atopic dermatitis, caused by general activation of immune-related signaling pathways. CC2, characterized by low expression of *PER* genes and *BMAL1*, had the youngest median age and a low prevalence of allergic rhinitis, a condition that is associated with decreased activation of the FcεRI signaling pathway. CC3, characterized by high expression of *BMAL1* and *CRY1*, had a low prevalence of atopic dermatitis, which is associated with decreased activation of cytokine–cytokine receptor interaction pathways. This is the first report to consider the expression of circadian clock genes when subcategorizing patients with asthma and explore the association between clinical features and asthma-related signaling pathways in each subtype.

We found that core circadian clock gene expression in pediatric/adolescent patients with asthma could be divided into 3 subtypes (CC1, CC2, and CC3). Interestingly, these subtypes showed distinct comorbidities and transcriptomic expression. In general, pediatric/adolescent asthma often co-exists with allergic rhinitis and atopic dermatitis, which was confirmed in the present asthma cohort. This may reflect an “allergic march” in children’s growth period [11]. We found a significantly low frequency of allergic rhinitis in CC2, and of atopic dermatitis in CC3. Further analysis of pathways in the KEGG database revealed changes related to this observation. In CC1, there was general activation of immune-related signaling pathways. In CC2, activation of the FcεRI signaling pathway was significantly decreased compared to that in CC1 and CC3. Because IgE sensitization is a crucial event in the development of allergic rhinitis [12,13], this finding might explain the low frequency of allergic rhinitis in CC2. CC3 was characterized by significantly decreased activation of cytokine–cytokine receptor interaction pathways compared with those of the other subtypes. This finding might explain the low frequency of atopic dermatitis in CC3, because there is accumulating evidence that activation of the IL-4/IL-13 signaling pathway is crucial for the development of atopic dermatitis [14,15]. However, because there were many differences in the activation of immune-related signaling pathways between the three CC subtypes, it remains to be determined which signaling pathways contribute to the distinct comorbidities observed in these groups.

It should be noted that the blood transcriptome is dynamic and has been reported to be altered by biological factors such as sex [16], age [17,18], body mass index (BMI) [19], and circadian rhythm [20]. There was no difference in the distribution of gender between the three CC subtypes in this study. However, CC2 had the youngest median age (4 years, compared to 16.8 years in both CC1 and CC3), which may explain the transcriptome distribution in CC2. We did not have information on BMI, which is a limitation of our study. Thus, the differences in comorbidity and transcriptomic expression could result from the asthma CC subtypes or from biological factors such as age or BMI.

In our analysis, among the seven core clock genes, we found only PER2 and PER3 have a slightly but significantly higher expression in the blood samples from asthma patients. In line with our results, PER2 and PER3 were increased, most noticeably when collected between 4 pm and 9 pm for PER2 and between 9 am and 9 pm for PER3, in peripheral blood mononuclear cells from asthma patients (unpublished results, scientific meeting abstracts [21,22]). In contrast, a recent study found that PER2 and PER3, together with other core clock genes, except for BMAL1, were downregulated in bronchial asthma patients [23]. A possible explanation for the different observations is the age of the research population. In our analysis, the database is from a young population, while Chen et al. [23] focused on an older population (50.98 ± 13.07 years and 50.63 ± 12.07 years in healthy and asthma patients, respectively). There are a few lines of evidence, although not in blood samples, that CCGs and circadian-controlled genes are differently expressed in different age groups across species [24,25,26]. The relationship between the circadian clock and the immune response is well reported. Core clock genes, individually, were linked to important immune and allergic functions. For example, BMAL1 deficiency caused an increase in the proinflammatory cytokines IL-1β and IL-6 in macrophages [27,28] and lymphocyte dysfunction [29,30]. Clock and Per2 were reported to control IgE- and IL-33-induced mast cell activation in mice [31,32,33]. Cry1 and Cry2 knockout fibroblasts increased NF–κB activation and expression of IL-6, TNF-α, and iNOS [34].

Although it does not belong to the seven core clock genes, the timeless circadian regulator TIMELESS has been reported to be associated with childhood asthma [35]. We found that TIMELESS expression in the blood was lower in the asthma group compared to normal patients (Appendix A), which was in concordance with previous research on asthmatic children [35]. Among the three asthma CCs, TIMELESS was slightly but significantly higher in CC2 (Appendix A). However, there was no association between TIMELESS and comorbidity (Appendix A).

An important question is whether the alterations in the transcription of core circadian clock genes in asthma patients are the cause or a consequence of the disease. Several rodent and human studies have suggested a causal role for the circadian clock in asthma. For instance, BMAL1 deficiency worsened viral bronchiolitis and promoted asthma-like airway changes [36]. *Rev-erbα* was reported to control circadian changes in the airway response in mice that were intranasally challenged with house dust mite allergen [4]. A recent long-term monitoring study in patients with asthma proved that pulmonary function was worst at night, even in healthy controls [3]. Interestingly, the risk of asthma in “owl” chronotype adolescents (who go to bed and wake up late) was 2.67 times higher than in “lark” chronotype adolescents (who go to bed and wake up early) [37]. On the other hand, sleep disorders are common in children with asthma [38] because asthma can interfere with the circadian rhythm. In addition, blood samples obtained from a healthy cohort showed that cortisol treatment altered the expression of *PER1* and *PER3* genes within an hour [39]. Thus, there seems to be a reciprocal relationship between circadian rhythm and asthma. Further studies are needed to advance our understanding of the relationship between altered circadian rhythm and asthma.

There are several limitations to this study. First, the transcriptomic data of blood cells (PBMCs) may not reflect those in the lung. Second, we had no clinical information except for that in the published database pertaining to asthma comorbidity with allergic rhinitis and atopic dermatitis [10]. Although the chance of other comorbidities was low, partially due to the age of the participants, they cannot be fully excluded. Lastly, because the time points at which blood samples were obtained were unclear in the published database [10], we cannot exclude the possibility that sampling times influenced the expression of circadian clock genes. Therefore, further studies considering this selection bias are needed to confirm our findings.

In summary, we found that, based on the core circadian gene expression patterns, pediatric/adolescent patients with asthma could be subdivided into three CC subtypes with distinct comorbidities and transcriptomic expressions. Although there are obvious limitations to this study, these findings suggest specific contributions of circadian clock activity to the pathophysiology of allergic diseases.

## 4. Materials and Methods

### 4.1. Data Processing

We accessed BioProject GSE141661 of the National Center for Biotechnology Information (NCBI) database to retrieve the gene expression and clinical data of patients with asthma [10]. There were a total of 457 blood samples corresponding to 457 patients. We excluded patients with only non-asthma diseases (*n* = 114) and included healthy individuals (*n* = 209) and patients with asthma (*n* = 134) as control and asthma cohorts, respectively. Reported clinical characteristics consisted of age, gender, allergy, asthma, dermatitis, and rhinitis.

### 4.2. Clustering Analysis

First, we applied a clustering algorithm using 7 core clock-controlled genes, including *CLOCK*, *BMAL1*, *PER1*, *PER2*, *PER3*, *CRY1*, and *CRY2*, in the asthma cohort. A 7-gene matrix was extracted from each sample. The expression of each gene across all included samples was scaled to a mean of 0 and a standard deviation of 1. Next, we applied K-mean clustering algorithms and calculated the WSS (elbow method), average silhouette width, and gap statistics with K values ranging from 1 to 10. We selected the K value based on these 3 metrics. Finally, the K value achieving the greatest consensus among the 3 selection methods was considered optimal.

### 4.3. Pathway Analysis

We performed GSVA, which is a variant of single-sample gene set enrichment analysis [40,41]. The enrichment score for each pathway was calculated in each sample. We then conducted ANOVA and Bonferroni corrections to compare the enrichment score of each pathway between CC clusters and reduce false-positive results.

### 4.4. Analysis Platform

The descriptive statistics of continuous and categorical variables were the median (range) and the number of patients (percentage), respectively. Wilcoxon and chi-square tests were performed to compare clinical variables between cohorts. All analyses were performed on R version 4.2.2 (The R Foundation, Vienna, Austria, URL https://www.R-project.org/).

## Figures and Tables

**Figure 1 ijms-24-07477-f001:**
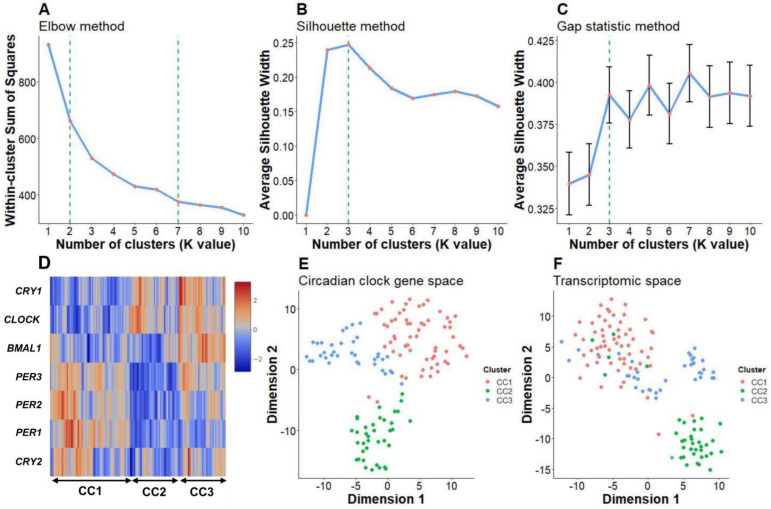
(**A**) The elbow method showing significant drops in WSS from K = 2 to K = 7, while the silhouette and gap statistic methods suggest that K = 3 is the optimal value (**B**,**C**), which leads to the consensus of K = 3 as the optimal number of CC clusters in patients with asthma. (**D**) Heatmap showing the expression pattern of the core clock genes; t-distributed stochastic neighbor embedding (t-SNE) dimension reduction illustrates that the three CC subtypes formed a distinct cluster in both circadian clock space (**E**) and whole transcriptomic space (**F**).

**Figure 2 ijms-24-07477-f002:**
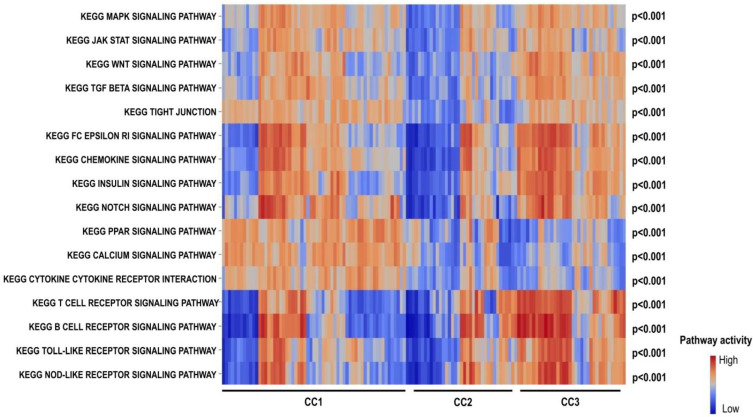
The heatmap demonstrates the activity of each pathway (row) within each patient (column). Samples within each cluster are grouped together, and each pathway is hierarchically clustered by its activity within the CC cluster. Pathway names are shown on the left, while the adjusted *p*-values of ANOVA tests with Bonferroni correction are shown on the right.

**Table 1 ijms-24-07477-t001:** Clinical characteristics of patients with pediatric/adolescent asthma.

			Asthma	
	Control (*n* = 209)	CC1 (*n* = 61)	CC2 (*n* = 37)	CC3 (*n* = 36)
Age	4.6 (3.8–18.1)	16.8 (3.9–18.0)	4.0 ** (0.1–17.0)	16.8 (4.5–17.8)
Gender				
Female	106 (50.7%)	28 (45.9%)	14 (37.8%)	15 (41.7%)
Male	103 (49.3%)	33 (54.1%)	23 (62.2%)	21 (58.3%)
Comorbidities				
AR		17 (27.9%)	3 (8.1%) **	16 (44.4%)
AD		14 (23.0%)	16 (43.2%)	2 (5.6%) **
AR & AD		11 (18.0%)	1 (2.7%)	4 (11.1%)
No		19 (31.1%)	17 (46%)	14 (38.9%)

** *p* < 0.01 compared to other asthma circadian clusters. AD, atopic dermatitis; AR, allergic rhinitis.

## Data Availability

The datasets analyzed for this study can be found in the National Center for Biotechnology Information (NCBI) database, BioProject GSE141661.

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
