# Peer review of "Association of Circadian Clock Gene Expression with Pediatric/Adolescent Asthma and Its Comorbidities"

_ijms, 2023, doi:10.3390/ijms24087477_

Round 1
Reviewer 1 Report
In the current manuscript, authors have presented an interesting subject of association of circadian clock gene expression with pediatric/adolescent asthma and its comorbidities. The studies consist multiple flaws in terms of study designs and interpretations. Specific comments are written below:
1. The characteristics of asthmatic patients at the time of evaluation is indispensable to include for eg., disease status, IgE levels or other features such as smoking, BMI, Obesity. Authors need to justify this.
2. It is not clear whether Asthmatic patients' data included in study were on medications or not. Are the asthmatic features of these patients were controlled or not?
3. Any exclusions of patients with other disorders such as any psychiatric problems or people working in night shifts or taking sleeping pills or any other chronic disease?
4. Apart from the mentioned circadian clock genes, Tim has been well studied and associated with asthma. Authors should have included Tim to correlate with other allergic comorbidities.
5. What is the baseline of circadian rhythm gene in the current studies? Any particular reason for not taking healthy subjects in consideration?
6. Discussion part can be strengthened with addition of immunological function of circadian genes in allergic conditions.
Author Response
Reviewer 1:
In the current manuscript, authors have presented an interesting subject of association of circadian clock gene expression with pediatric/adolescent asthma and its comorbidities. The studies consist multiple flaws in terms of study designs and interpretations. Specific comments are written below:
We thank the reviewer for the precious time carefully reviewing our manuscript and providing valuable comments. It was your valuable and insightful comments that led to possible improvements in the current version. The authors have carefully considered the comments and tried our best to address every one of them. There were several comments that the authors cannot fully address due to the lack of information in the original database. Please refer to the detailed answers below:
Comment 1: The characteristics of asthmatic patients at the time of evaluation is indispensable to include for eg., disease status, IgE levels or other features such as smoking, BMI, Obesity. Authors need to justify this.
Response: We appreciate the reviewer’s insightful suggestion that clinical characteristics of asthmatic patients were important aspects to consider. However, the information was neither available from the publicly available datasets nor mentioned in the original article where this dataset was used (Please refer to Lemonnier, et al., Allergy. 2020; 75: 3248– 3260. https://doi.org/10.1111/all.14314). This is a limitation of database-based study, especially in the allergic field. We hope that in near future, there will be a comprehensive database available for the allergic diseases.
Comment 2: It is not clear whether Asthmatic patients' data included in study were on medications or not. Are the asthmatic features of these patients were controlled or not?
Response: We agreed that the association of asthma control status and circadian clock is of great interest. However, medication and control status information were not available from the original article, as they focus on allergic comorbidities. As a results, the published datasets (National Center for Biotechnology Information (NCBI) database bio project GSE141661) was only available with limited clinical characteristics (age, gender, allergy, asthma, dermatitis, and rhinitis). We used all the available information in our analysis.
Comment 3: Any exclusions of patients with other disorders such as any psychiatric problems or people working in night shifts or taking sleeping pills or any other chronic disease?
Response: Thank you for your question. From the article which the datasets for this study were extracted, the authors did not mention about any other psychiatric disorders or other chronic disease except for asthma, dermatitis, and rhinitis. We tried our best to trace back to the original cohort. The BAMSE and part of the INMA cohort included in the datasets was focus on environments and allergy disease (please refer to https://ki.se/en/imm/bamse-project for the BAMSE cohort and Mònica Guxens et al., International Journal of Epidemiology, Volume 41, Issue 4, August 2012, Pages 930–940, https://doi.org/10.1093/ije/dyr054 for the INMA cohort). In our analysis, we exclude patients with non-asthmatic disease (Material and Method, data processing, line 266 to 268). As the database was from child/adolescent cohort, we think that night shift was irrelevant. We discussed that sleeping problem in asthma patient may play some roles in the different expression of core clock genes (Discussions, line 240 to 242).
Comment 4: Apart from the mentioned circadian clock genes, Tim has been well studied and associated with asthma. Authors should have included Tim to correlate with other allergic comorbidities.
Response: We thank the reviewer for providing suggestion to include TIMELESS. We did not include TIMLESS in the genes used for clustering asthma patients as we aimed to focus on the core circadian clock genes of the classical translational-transcriptional feedback loop. Taken that TIMELESS would have important association with childhood asthma (Langwinski W et al., Clin Respir J. 2020 Dec;14(12):1191-1200. doi: 10.1111/crj.13260), we performed a comparison for TIMELESS expression between asthma and normal patient. TIMELESS expression was lower in asthma group compared to normal patients. Previous research reported a decreased expression of TIMELESS in the blood of asthmatic children. We further analyzed TIMELESS expression between circadian cluster of asthma patients and examined TIMELESS expression with comorbidity. We mentioned these finding in Discussion part (line 223 to line 230): “Although not belongs to the 7 core clock genes, TIMELESS (timeless circadian regulator) has been reported to associated with childhood asthma [35]. We found that TIMELESS expression in the blood was lower in asthma group compared to normal patients (Supplementary Figure 2. A), which was in concordance with previous re-search of asthmatic children (https://pubmed.ncbi.nlm.nih.gov/32790948/). Among the 3 asthma CCs, TIMELESS was slightly but significantly higher in CC2 (Supplementary Figure 2. B). However, there was no association between TIMELESS and comorbidity (Supplementary Figure 2. C).”
Comment 5: What is the baseline of circadian rhythm gene in the current studies? Any particular reason for not taking healthy subjects in consideration?
Response: Thank you for the nice question. The expressions of core clock genes in our analysis were scaled and relatively compared within asthma patients (Materials and Methods, Clustering analysis, line 272 to 279). We included the healthy subjects for determining the different expression of these core clock genes between asthma patients and normal patient. We did not include that result in the first submission for two reasons: (1) our study focused on the relationship between core circadian clock genes in the pathophysiology of asthma, and (2) normal patients had differed core clock gene clusters from asthma subjects (Supplementary Fig. 2). These clusters were not related with the disease and just an observation. We took this comment into account, together with comment #2 of reviewer 2, in this reviewed submission, we included the result from normal patients in supplementary figure 1 and 2. We discussed about the different expressions of the core circadian clock genes in the Discussion part (line 204 to line 216).
Comment 6: Discussion part can be strengthened with addition of immunological function of circadian genes in allergic conditions.
Response: We appreciate the reviewer’s suggestion for improving the discussion part. This suggestion even becomes more useful when we included the different expression of core clock gene between control and asthma cohort. We added a few line in the discussion part, together with the discussion of the different expression between control and asthma cohort (line 216 to 222): “The relationship between the circadian clock and the immune response is well-reported. Core clock genes, individually, were linked to important immune and allergic functions. For example, BMAL1 deficiency caused an increase of proinflammatory cytokines IL-1β and IL-6 in macrophages [27,28], and lymphocyte dysfunction [29,30]. Clock and Per2 were reported to control IgE- and IL-33-induced mast cell activation in mice [31-33]. Cry1 and Cry2 knockout fibroblasts increased NF–κB activation and expression of IL-6, TNF-α, and iNOS [34].”
Reviewer 2 Report
The article ‘Association of circadian clock gene expression with pediatric/adolescent asthma and its comorbidities’ submitted by Tran NQV and Le MK investigates the relationship between the expression of circadian clock genes with clinical features of pediatric/adolescent asthma. Authors have analyzed the published transcriptomics data of peripheral blood mononuclear cells from individuals with pediatric/adolescent asthma and classified these patients into three categories based on the expression pattern of seven core circadian genes (CLOCK, BMAL1, PER1-3, CRY1-2). I have a few concerns with this manuscript.
1. My major concern with this manuscript is stated by the authors, too- ‘The time points at which the blood samples were obtained, were unclear.’ The RNA/Protein levels of circadian clock genes vary in accordance with the day-night period. That makes it necessary to collect the samples at the same time point. The variation in the expression of circadian clock genes could be due to the difference in sample collection time points.
2. Control group placebo samples are missing in the analysis. That makes it difficult to appreciate the findings. Comorbidities associated with CC2 could be due to the lower median age of the children since authors do not have control group placebo data.
Author Response
Reviewer 2
The article ‘Association of circadian clock gene expression with pediatric/adolescent asthma and its comorbidities’ submitted by Tran NQV and Le MK investigates the relationship between the expression of circadian clock genes with clinical features of pediatric/adolescent asthma. Authors have analyzed the published transcriptomics data of peripheral blood mononuclear cells from individuals with pediatric/adolescent asthma and classified these patients into three categories based on the expression pattern of seven core circadian genes (CLOCK, BMAL1, PER1-3, CRY1-2). I have a few concerns with this manuscript.
We thank the reviewer for the precious time carefully reviewing our manuscript and providing valuable comments. We agree with the reviewer concern, as it was also our concern as a limitation of this study. We appreciate the reviewer’s insightful suggestion and agree that it would be useful to include control group in our analysis, which may support our findings. Please refer to our responses below for details.
Comment 1: My major concern with this manuscript is stated by the authors, too- ‘The time points at which the blood samples were obtained, were unclear.’ The RNA/Protein levels of circadian clock genes vary in accordance with the day-night period. That makes it necessary to collect the samples at the same time point. The variation in the expression of circadian clock genes could be due to the difference in sample collection time points.
Response: We totally agree with the reviewer concern about time of blood collection. We traced back to the original BAMSE and INMA cohort studies, from which the transcriptomes were extracted, but we could not find any information about time of blood collection (please refer to https://ki.se/en/imm/bamse-project for BAMSE cohort and Mònica Guxens et al., International Journal of Epidemiology, Volume 41, Issue 4, August 2012, Pages 930–940, https://doi.org/10.1093/ije/dyr054 for the INMA cohort. Nevertheless, we acknowledged this concern as our limitation in the current study.
Comment 2: Control group placebo samples are missing in the analysis. That makes it difficult to appreciate the findings. Comorbidities associated with CC2 could be due to the lower median age of the children since authors do not have control group placebo data.
Response: Reviewer 1 also comment on the important of control group. We appreciate both reviewers for this suggestion to improve and support our findings. We found that control groups could be separate into 2 circadian clusters with distinct transcriptomic profile. However, further analysis is beyond the scope of our paper, which aims only to show that circadian clusters in the asthma groups were associated with the comorbidities. We also mentioned these finding in result 2.2 (line 101 to line 103): “Applying the same analysis workflow for healthy control group resulted in 2 distinguished CCs (Supplementary Figure 2), suggesting that the 3 CC subtypes were characterized for asthma patient.” and result 2.3 (line 118 to line 122): “These observations also found in the control groups (Supplementary Figure 2E and F), suggesting the biological differences between individual may play important roles in shaping the transcriptomic profile. However, further exploring the relationship between biological background, circadian clock, and transcriptomic profile is out of scope of this study.”
Round 2
Reviewer 1 Report
Authors have justified all my comments and I have no further concerns.
Reviewer 2 Report
Although authors have addressed my concerns but research design for this study is not appropriate as one cannot study the circadian clock without without considering the sample collection time points.